# A3C System: One-Stop Automated Encrypted Traffic Labeled Sample Collection, Construction and Correlation in Multi-Systems

Zihan Chen [1,2,3] , Guang Cheng [1,2,3,*], Ziheng Xu [1,2,†], Keya Xu [1,2,†], Yuhang Shan [1,2,†] and Jiakang Zhang [1,2]

1   School of Cyber Science and Engineering, Southeast University, Nanjing 211189, China
2   Jiangsu Province Engineering Research Center of Security for Ubiquitous Network, Nanjing 211189, China
3   Purple Mountain Laboratories, Nanjing 211189, China
*   Correspondence: chengguang@seu.edu.cn
†   These authors contributed equally to this work.

**Featured Application: The results of this paper can be used in all the fields related to encrypted traffic analysis, especially in the field of encrypted traffic classification and encrypted traffic identification which depend on a large number of labeled samples.**

**Abstract:** Encrypted traffic classification can essentially support network QoS (Quality of Service) and user QoE (Quality of Experience). However, as a typical supervised learning problem, it requires sufficiently labeled samples, which should be frequently updated. The current gateway-based labeled sample acquisition methods can only be carried out under TLS traffic. It relies on the Server Name Indication, a confused optional field that can be tampered with. The current end-based methods carried out manually or automatically have low efficiency and lack sample integrity, category purity, and label authenticity. In addition, they may have colossal packet loss and violate device security and user privacy. To solve these problems, we propose a one-stop automated encrypted traffic labeled sample collection, construction, and correlation system, A3C. First, we carry out the automated process-isolated traffic collection and labeled sample construction in the mixed application scenario, which can be used on Windows, Linux, and Android systems. Then, we propose the Segmented Entropy Distribution Capsule Neural Network (SED-CapsNet) to validate the encryption of the collected samples. We also propose optional authenticity validation and context flow correlation methods. Experimental results show that the system can effectively achieve one-stop encrypted traffic labeled dataset acquisition. It is superior to the existing methods.

**Keywords:** network security; encrypted traffic analysis; automated labeled traffic sample collection; multi-systems; encryption detection; SED-CapsNet

## 1. Introduction

The number of Over-The-Top (OTT) applications and services increases rapidly with technology development, enriching people's lives. Network traffic is the direct manifestation of OTT application data, and it is the data that ISPs can legally and transparently access with privacy protection. With the rise of user privacy protection and network security awareness, encryption technology has been widely used in network communication. Google's Transparency Report "Encrypted traffics in all Google products and services" [1] presented that almost all the traffic of Google products and services is 100% encrypted in October 2022, except for Google Maps (99% encrypted) and Google News (95% encrypted). Nevertheless, while encryption brings privacy and security to enterprises and users, it also challenges network management and security.

As a form of expression rich in data and metadata, information needs to be obtained through analysis. Based on traffic analysis, ISPs can effectively optimize network QoS

(Quality of Service) or user QoE (Quality of Experience) by network management, such as online video traffic acceleration, and even detect the cyber threats in advance. The significance of encrypted traffic classification is shown in Figure 1.

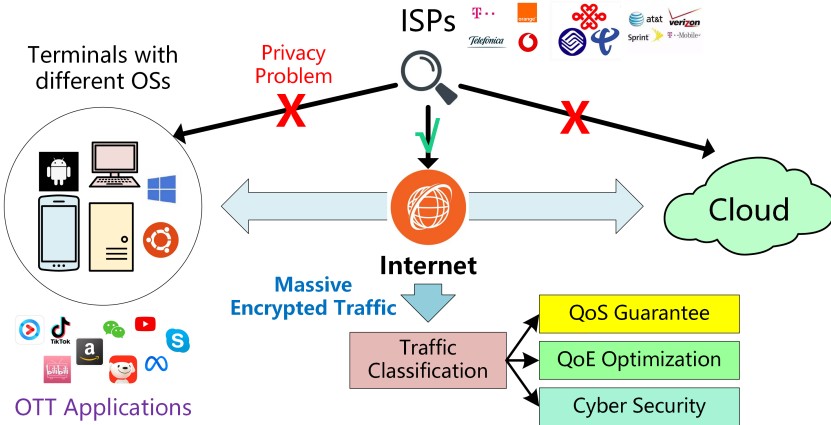

**Figure 1.** Significance of encrypted traffic classification.

Encryption covers plaintext information that can be obtained by simple means such as matching, significantly reducing the disclosure of traffic information. It also protects malicious software that wants to hide its whereabouts, making the original Deep Packet Inspection (DPI)-based [2] encrypted traffic classification method invalid. This phenomenon makes making it a critical research field. The problem is mainly presented in Figure 2.

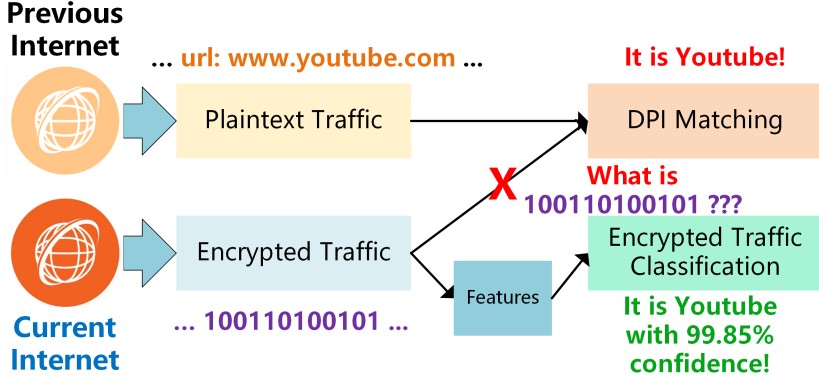

**Figure 2.** Problem of network traffic encryption.

As a typical supervised learning problem, encrypted traffic classification must process with labeled datasets for model training.

Many researchers use public datasets to train their model. It is undeniable that public datasets, such as ISCX VPN-nonVPN [3], have been widely used with its authority. However, the current public datasets are relatively old, and there are wide differences between them and the current network traffics, which makes it difficult for the trained model to apply to the current network. The significant differences are protocols, and the most representative ones are TLS-1.3, encrypted Domain Name System (DNS), and QUIC [4]. The new encryption protocols are rapidly iterating and upgrading, and combined with the endless proliferation of proprietary protocols together significantly reduce the availability period, continuously bringing conceptual drift in features. Some researchers hope to collect and construct private datasets with better timeliness by themselves manually. However, due to the lack of unified construction standards and the possibilities of privacy leakages, private datasets are challenging for others to obtain for experimental verification, leading to the lack of recognition.

The existing network traffic labeled sample collection is mainly divided into two types. One is executed on the gateway (or through the network splitting), and the other is carried out on the end system. The main difference lies in the labeling. The labeling on the gateway is mainly carried out by using the plaintext information remaining in the flow, among which the Server Name Indicator (SNI) based on the TLS header is the most representative one. However, SNI exists only in TLS. As a result, UDP and some non-TLS TCP traffic cannot be labeled. In addition, as an extended field, SNI does not necessarily exist and can be tampered with, resulting in labeling errors. Furthermore, the experiment found that as SNI is a domain name segment, multiple applications of one company may use the same SNI, and applications of different companies may call each other's services, resulting in further confusion about SNI. Eventually, with Encrypted SNI (ESNI) and TLS-1.3, SNI will become obsolete.

Manual collection and labeling methods based on the traffic sniffing tool (such as Wireshark [5]) are much less efficient than gateway traffic filtering but they has better precision. However, there are still many problems with manual methods. Firstly, there is a manual error in this process, which lies in the integrity of samples and the correctness of labels. Secondly, to ensure the correctness of labels, it is necessary to ensure that there is only one target application that needs to be collected and labeled in the current operating system (OS) environment. This process is difficult to achieve, and there is interference from the background traffic of the OS. Third, in some OSs, such as Android, traffic collection and labeling require root authority, resulting in privacy and security risks. To break this situation, some attempts have been made to automate the collection of labels, such as OpenQPA [6] and PacketCapture [7]. OpenQPA has the problem of packet loss, and the system is not open source, resulting in incomplete, unusable, and unreliable samples. However, PacketCapture requires the installation of user certificates, resulting in privacy and security risks. In addition, the obtained samples are HTTP information assembled after decryption, which cannot be used to encrypt traffic classification.

Overall, there are seven difficulties to overcome: (1) execute on the end system, supporting a variety of OSs, with satisfactory efficiency; (2) need to obtain the correct application labels and automatically label the samples; (3) need to include user behavior, that is, can operate the application manually, but can not introduce manual error; (4) the sample needs to be complete, with context, without packet loss; (5) users' privacy shall be protected, device security shall not be damaged, and keys shall not be obtained for decryption; (6) purity of samples shall be ensured, and background traffic of the OS shall not be mixed; and (7) collection and labeling procedure shall not affect the features of traffic, and the regular use of users shall not be affected.

Hence, we propose a one-stop architecture named automated encrypted traffic labeled sample collection, construction, and correlation (A3C). Firstly, we analyze the principle and form of encrypted traffic and decide the end's automated encrypted traffic sample collection. This architecture includes the online acquisition procedure and the offline validation procedure. The main reason for the asynchronous design is that the validation process is complex and optional, requiring a considerable computation cost compared to the online acquisition procedure. Then, we adopt semi-auto and fully-auto modes to collect traffic. The differences lie in the relevance between traffic and user behavior. On this basis, we realized cross-platform process-level traffic collection and sample construction for three different OSs, respectively, by utilizing the different interfaces. Afterward, the labeled samples of multiple processes of complex multi-process applications are correlated, and process-level samples are combined into application-level samples. In the offline validation procedure, we can conduct encryption detection on traffic samples to purify labeled datasets. Encrypted traffic using known encryption protocols can be directly screened, and the encryption of unknown protocols was detected by the proposed Segmented Entropy Distribution Capsule Neural Network (SED-CapsNet). After that, TLS parameter validation is implemented to verify standard TLS encrypted traffic to ensure the sample authenticity. Finally, the consistency between DNS and HTTPS parameters is used to correlate DNS

traffic samples with HTTPS traffic samples to guarantee the integrity of labeled samples. The system solves seven difficulties in one go and it has been released at: https://data. iptas.edu.cn/web/tbps (accessed on 15 April 2022). An anonymized dataset obtained by A3C has also been published at: https://data.iptas.edu.cn/web/tbps (accessed on 15 April 2022).

Our main contributions can be summarized as follows:

- We point out that the most serious bottleneck in the field of encrypted traffic classification currently lies in reliable and continuously updated datasets. Then we describe the problems existing in encrypted traffic labeled sample acquisition. To solve the core labeled sample acquisition problems, we break the research threshold of large-scale labeled dataset requirements by putting forward the A3C architecture.
- We develop A3C into three system instances on Windows, Linux, and Android. We break through the limitation of the traditional traffic sample collection method, which needs to restrict the currently running application. We realize the correspondence between the process and traffic on different OSs to carry out the automated process-level isolated collection and labeled sample construction of the traffic in the mixed application scenario.
- We propose a sample encryption validation method based on SED-CapsNet to detect the encryption of the collected data and realize the purification of samples. Our method performs much faster than the state-of-the-art HEDGE method and can still achieve high effectiveness (1/8 time cost with 83.12% F1-score).
- We make public the A3C system and an anonymized dataset obtained through the system under an existing large-scale network, which break the current situation that the research on encrypted traffic classification requires a large-scale experimental environment and considerable labor cost.

The rest of the paper is structured as follows. Section 2 analyzes the commonly used datasets, dataset construction methods, and classification methods of encrypted traffic. Section 3 briefly introduces the encrypted traffic classification and the encrypted traffic sample collection. Section 4 presents the whole picture of the A3C system. Then it describes the methods used in the online acquisition procedure. Section 5 describes how to validate the encrypted flow. In Section 6, the A3C system is evaluated through experiments in many aspects. Finally, we summarize our research and prospect the future.

## 2. Related Work

The basis of encrypted traffic classification lies in the labeled datasets, and the core lies in the features and classification model. Therefore, we first summarize and analyze the current datasets and dataset construction methods in encrypted traffic analysis. Then we introduce some research progress in encrypted traffic classification.

### 2.1. Encrypted Traffic Datasets and Construction Methods

2.1.1. Classic Open Datasets in Encrypted Traffic Analysis

The datasets of encrypted traffic analysis can be traced back to malicious traffic analysis. As one of the most classic datasets, DARPA99 [8] is an intrusion detection dataset which contains Secure Shell (SSH) traffic [9]. The datasets of detection currently include the University of MASSachusetts (UMASS) dataset [10], the University of California Irvine Security (UCIS) dataset [11], the MAWILab (MAWI) dataset [12], and the National Incident Management System (NIMS) dataset [13]. In encrypted malicious traffic detection, there are the Honeynet Project ISOT(ISOT) dataset [14], the MCFP Project dataset [15], and the University of Science and Technology of China TFC (USTC-TFC) dataset [16]. They all have some encrypted traffic samples.

The most popular dataset in encrypted traffic classification is the ISCXFlowMeter based (ISCX) VPN-nonVPN dataset [3] published in 2016. It provided essential support for various classification researches. However, it is worth noting that not all encrypted traffic datasets contain malicious traffic, which mainly depends on the purpose of the dataset. It

contains malicious data if the dataset is used for malicious traffic classification or intrusion detection. If it is for application classification, especially for network management-oriented research, the dataset contains only the benign traffic of multiple applications or services.

It is easy for subsequent researchers to validate the effect of their research because the open datasets have been widely used in many pieces of research. However, through in-depth analysis of the above datasets, it can be found that there is a severe problem of unbalanced sample distribution, and the encryption protocol and application protocol versions are also out of date to some extent. The above defects make it difficult for the open datasets to represent the traffic of the current network environment.

### 2.1.2. Private Datasets from Existing Researches

Given that many defects of the current public datasets exist, some research institutes that can build labeled datasets begin to use private datasets to conduct research.

Such datasets include private ones collected from top N websites published on the Alexas [17,18], self-collected datasets from CERNET [19], from ISP [20], datasets generated from experimental environments [21], and datasets generated by global mobile solution providers [22].

However, the private datasets are not open for other researchers to use, and even some researchers do not provide the details of the datasets in their papers, which leads to the lack of credibility. At present, the construction of datasets is mainly carried out by traffic analysis tools combined with manual collecting and labeling behaviors. As a result, there is an interference of background traffic from the OS and possible manual errors. In addition, if the private datasets are not collected in a large-scale actual network environment, the datasets will have a considerable concept drift so that the trained model is not universal.

### 2.1.3. Dataset Construction and Improvement Methods

Due to the small and unbalanced size of the existing open datasets, some researchers focus on improving the datasets, not the data collection stage.

Iliyasu A S et al. [23] proposed a semi-supervised learning method using a Generative Adversarial Network (GAN) combined with deep learning to solve an unbalanced dataset problem, called DCGAN. Unlike DCGAN, Wang Z X et al. [24] directly used GAN to fill unbalanced encrypted traffic samples. Then, Wang P et al. [25] proposed PacketCGAN, based on the conditional GAN. In addition, Zheng W et al. [26] proposed a method through a "hallucinator" neural network for the problem of unknown datasets.

GAN can fill the encrypted traffic datasets, but it cannot reach the new encryption sample with different types. It also cannot add other characteristics within the exact category characterization of the samples.

In addition to GAN, there are currently several traffic generation tools, such as App-Scanner [27], Lashkari [28], and ExperimenTor [29]. The above tools need to run in the experimental environment, which can only ensure that the protocol features of traffic are similar to that of the actual traffic. The lack of user behavior and environment characteristics makes the samples lack authenticity and reliability.

### 2.2. Encrypted Traffic Classification and Feature Engineering

### 2.2.1. Encrypted Traffic Detection

Encrypted traffic detection is a special field in encrypted traffic classification, and its purpose is to screen encrypted traffic from network traffic and provide a guarantee for the input of encrypted traffic classification. Although plaintext traffic can be regarded as encrypted traffic in actual classification, in the training process, the significance of the features of plaintext traffic may seriously interfere with the category common feature learning of the encrypted traffic classifier, resulting in the appearance of overfitting.

At present, the randomness measure is used to verify the encryption of traffic. Dorfinger et al. [30] proposed a method for encrypted traffic detection by splitting packets into byte units and taking advantage of the difference in coverage interval and distribution between

encrypted and unencrypted traffic byte units in the ASCII encoding field. Zhao et al. [31] proposed EIWCT, a time-delay adaptive encryption traffic blind recognition algorithm based on the weighted cumulative sums test, to solve the problem that entropy measure is difficult to fine control the collection time and it ignores the random variation trend of data. However, as a core network carrying various other forms of data transmission, the Internet has some compressed traffic to ensure network QoS and energy consumption of some devices. In data processing, data compression and encryption both increase the information entropy of the source data [30]. If random measures detect the encrypted traffic without considering the existence of compressed traffic, the result may be interfered with by compressed traffic, and the false positive rate would be high. In recent years, the academic community has gradually considered the interference of compressed flow. Therefore, Casino et al. [32] proposed a method HEDGE based on multiple random measures to separate flow-transformed encrypted files, flow-transformed compression files, and flow-transformed plaintext files. The goal of HEDGE is to distinguish the encrypted, compressed, and plaintext traffic. However, its experiment is implemented by transmitting the encrypted file, the compressed file, and the plaintext file, respectively, in the TCP environment. There is some difference between an encrypted file and the actual encrypted traffic, and the difference mainly lies in the mechanism of the security protocol. Several random measures specified in NIST SP 800-22 are tested in this method. Finally, three random measures, frequency within block test, cumulative sums test, and approximate entropy test, are selected as the separation indexes. The encrypted traffic detection is realized effectively.

With the development of machine learning technologies, a hybrid method combining machine learning and random measures has been developed. Niu et al. [33] put forward the Heuristic Statistical Testing (HST). Traffic randomness was measured based on four random measures, and C4.5 was used for encrypted traffic detection.

Tang et al. [34] proposed an entropy-based feature extraction algorithm to solve the problem that encrypted traffic and unencrypted compressed traffic need to be distinguished before subsequent encrypted traffic classification. They converted the traffic into entropy vector features through a fixed-length sliding window to imply the classification of encrypted traffic and compressed traffic.

According to the comparison of experimental results of the above papers, the most representative research in encrypted traffic detection is HEDGE.

### 2.2.2. Machine Learning-Based Models

There are a lot of traditional machine learning methods, and most of them have been used in encrypted traffic classification. The most used ones are ensemble learning methods, which perform a better effect. There are integrated Markov chain classifiers WENC [20], gradient boosting tree [35] and RF [36]. After that, there are also some optimized methods such as the quadratic voting RF [37]. However, Fathi-Kazerooni S et al. [38] proposed a traffic camouflage method, effectively protecting user traffic from recognition by RF. As a result, it may be difficult for traditional machine learning methods to classify encrypted traffic in the future.

### 2.2.3. Deep Learning-Based Models

The most significant advantage of deep learning is that it can directly execute with no prior expert knowledge of feature engineering. When used in encrypted classification, these models can directly take the raw encrypted traffic as the input. The base model includes CNN [39], text-CNN [40], LSTM [41] and CapsNet [42]. With the development in clustering methods, the combination of different deep learning models makes a better effect than single model, such as Deep Packets [43] (integrated CNN and SAE), STNN [28] (integrated LSTM and 3D-CNN), LS-CapsNet [44] (GRU for feature extraction and CapsNet for classification), and CENTIME [45] (integrated ResNet and AutoEncoder).

These deep learning methods improve the accuracy and lower the research barrier, but the performance is still not satisfied in the actual network environment.

### 2.2.4. Encrypted Traffic Feature Engineering

Although the deep learning method can automatically select features, it does not mean that the selected features are optimal. In addition, these features are not interpretable. Therefore, some researchers start from feature engineering to optimize deep learning in encrypted traffic classification as the automatically selected features cannot effectively extract complex high-dimensional Markov features.

In time series features, there are FlowPic [46] and the non-zero load packet FlowPic [47]. Then, Baldini G et al. [48] transformed the time-related features and took the transformation as classification input.

Some researchers attempt to make use of residual plaintext features, which could only be useful in some particular scenes, such as simultaneous queries of multiple hostnames on Android mobile terminals [49] and plaintext DNS traffic with handshake certificate [50].

Many studies have shown that the length features are strongly relevant to the transmission data. The representative studies include the SoB method [27] using the length of the certificate in TLS handshake and the first TLS application data size, FS-Net [21] based on the multi-layer bidirectional GRU model with representation learning, TFSN model [51] based on LSTM and composite deep learning model architecture LSCDL [19] with PDU length sequence as the input and N-gram length sequence as the features.

Both feature optimization and model iteration improve classification performance in the case of fixed datasets. However, datasets and features determine the upper bound of classification, and models and algorithms approximate it. Therefore, it is necessary to realize automated labeled sample construction and efficiently continuously sample updating, as the A3C proposed in this paper.

## 3. Preliminaries

### 3.1. Encrypted Traffic Classification

The aim of encrypted traffic classification is to classify the encrypted traffic into some specific categories which are applications or services at present, under the condition of determining the number and label of categories. Without considering the unknown applications or services, encrypted traffic classification is a typical supervised learning scenario.

For encrypted traffic classification, there are three elements to be quantified, which are sample, feature, and category. Assuming there are $n$ samples to be classified, $F$ is features used to input and $C$ is different categories. The samples $X$ can be described as $X = \{x_1, x_2, \cdots, x_n\}$, where $x_i$ refers to the $i$-th sample. The feature set can be described as $F = \{f_1, f_2, \cdots, f_m\}$, where $m$ represents the number of features in $F$ and $f_j$ refers to the $j$-th feature. An input sample $x_i = [f_1^{(i)}, f_2^{(i)}, \cdots, f_m^{(i)}]$ where $f_j^{(i)}$ is the $j$-th feature in $i$-th sample. Taking the application classification as an example, if the real category of $x_i$ is $a_i$, $a_i = c_k, c_k \in C$, the goal of the encrypted traffic application classification is to build a model $\phi(x_i)$ to obtain a predicted label $\hat{a}_i$ which is expected to be the real label $a_i$.

### 3.2. Encrypted Traffic Sample Collecting and Labeling

If we need to train a encrypted traffic classification model, we need a labeled dataset $L$ contains both samples $X$ and the corresponding categories of each sample $A$. Hence, $L = \{(X, A)\} = \{(x_1, a_1), (x_2, a_2), \cdots, (x_n, a_n)\}$. It is easy to obtain $X$, but hard to acquire the corresponding $A$ at the same time with integrity, purity, authenticity, and credibility.

The label and the sample should be acquired simultaneously because, in encrypted traffic communication, the label is the source of the sample (application generates the traffic). The integrity means that the samples in a category need to cover as many application scenarios as possible and be complete, while the total category of samples also needs to meet the objectives of the classifier, which is $set(A) \subseteq C$. Purity means that there should be no other types of samples and interference traffic led by background traffic in a certain sample,

$\forall x_p, x_q \in X^{C_k}, a_p = a_q$. No category is not in $C$, which is $C \subseteq set(A)$; hence, $set(A) = C$. Authenticity means that labeled samples meet the needs of classifiers through verification. Credibility means the process of sample collection and labeling must be credible.

## 4. Automated Online Traffic Collection, Sample Construction, and Correlation

In this section, we give an overview of the A3C system. Then, we focus on three of four core elements: credibility, integrity, and purity, and propose an automated online encrypted traffic collection, sample construction, and correlation method. It is the underlying primitive sample support of the A3C system.

### 4.1. System Overview

The classification of encrypted traffic is similar to other supervised learning classification problems in machine learning, including training and applying. The labeled datasets are used in the training phase. It is worth noting that the sample collection should be deployed on the end because almost all the traffic is encrypted in the communication link for ISPs. Even the deep packet inspection devices cannot determine the application of TLS traffic since the plaintext parameter, like SNI, is changeable. On the contrary, the data are generated on ends and servers, and the user behaviors are sourced from the applications on the end. The data are not encrypted before becoming the traffic. Therefore, in the end system, we can make sure what the traffic is and collect the traffic simultaneously.

To guarantee the integrity, purity, authenticity, and credibility of the dataset, we propose the A3Cautomated encrypted traffic labeled sample collection, construction, and correlation architecture, as shown in Figure 3. The A3C includes two procedures: the online acquisition procedure and the offline validation procedure, in which there are five modules. The asynchronous scheme can significantly reduce the overhead of collecting devices because the sample acquisition is time-sensitive, but validation is not. Validation requires a much higher calculation cost because of the high complexity computation, especially in encryption detection. Each module in Figure 3 is described in detail later.

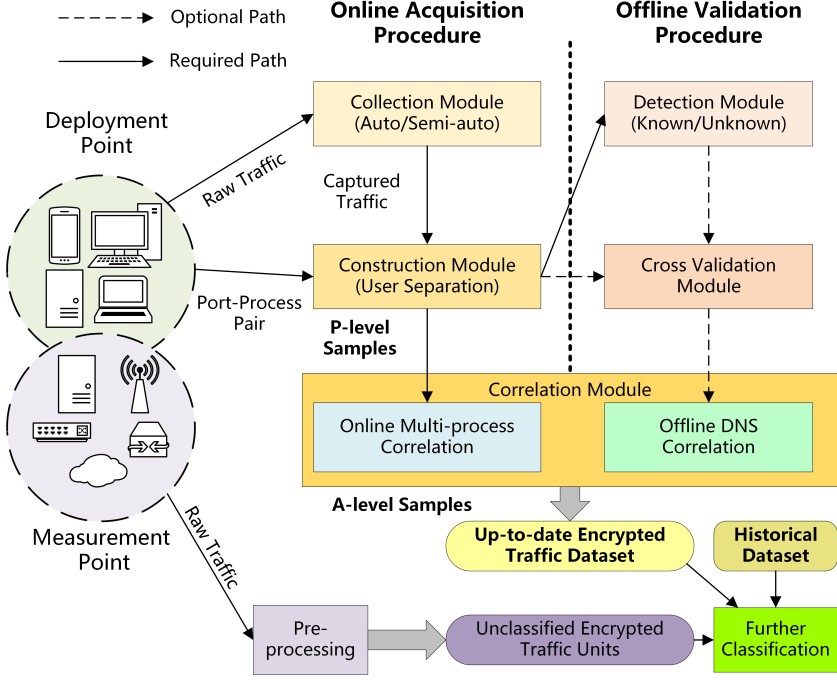

**Figure 3.** A3C system architecture.

Although we have deconstructed the entire lifecycle of automated encrypted traffic labeled sample acquisition and designed the A3C system architecture, many issues still need further study. The first is the operating system platform. The collection and construction

of encrypted traffic samples cannot be separated from the end system. If encrypted traffic sample data with broader coverage and various environments are required, it is necessary to make A3C capable of supporting various mainstream OS platforms.

The second problem is the interference between background traffic and other applications. The current traffic sample collection scheme needs to limit the application's network access, including a particular pre-operation cost and OS background traffic interference. In an OS, the running unit of the application is the process. If we want to separate the traffic belonging to different applications, we need to separate the traffic at the process-level first.

The third is the indeterminate coupling between user behavior and application traffic. That is, the traffic generated by the application is either spontaneous or user-generated, which needs to be considered.

### 4.2. Cross-Platform Traffic Collection and Sample Construction

The three leading OS platforms are Windows, Linux, and Android. Therefore, A3C must take into account the specificity of these three operating systems to realize cross-platform traffic collection and sample construction.

To solve the problem of sample purity, we delve into the mechanisms by which operating system processes generate network traffic. Processes have their own unique PID, and each process independently identifies its application traffic by the port number per unit of time. Therefore, we propose a cross-platform traffic collection and sample construction method based on the "process-PID-port-flow" graph.

Moreover, no matter what the OS is, the application generates traffic in two behaviors: user-relevant behaviors and user-irrelevant behaviors. User-relevant behaviors represent how users interact with the application, and the application generates traffic during the interaction. User-irrelevant behaviors are spontaneous of the application, which produces network traffic.

#### 4.2.1. Windows

The Windows platform has the most significant number of application types. However, its user affinity leads to a large number of resource-intensive applications. We first use the netstat command and network adapter packet capture to realize sample capture in a Windows environment and confirm the current packet's process by querying the "PID-port" pair returned by netstat. However, severe packet loss occurs because netstat polling consumes many system resources. In Windows, interactions are event-based.

The event will be recorded in the system, including the process of access to the network. These can be obtained through WinAPI system functions. Hence, we propose an A3C optimization method for the Windows platform using WinAPI. In WinAPI system functions, two functions can obtain the network access relationship of the process, respectively, namely *GetExtendedTcpTable* and *GetExtendedUdpTable*. These two functions can obtain the current TCP and UDP network connection status, including the port to PID mapping relationship, and do not need to read the disk. These two functions retrieve tables we call Extended TCP Table (ETT) and Extended UDP Table (EUT), respectively.

In addition, to obtain the relationship between the process name and PID, we need to obtain the running status of the current process in the system when the traffic is generated. This information can also be obtained through the WinAPI function *CreateToolhelp32Snapshot*, known as the snapshot of processes (SoP). A3C-Windows sample collection and construction procedure is demonstrated in Figure 4. It can be divided into three phases: traffic collection, packet parsing, and process mapping.

In order to realize automation and improve efficiency to prevent packet loss, we first automatically identify network adapters through the gateway and pre-initialize port-PID mapping (PP mapping), PID-process name mapping (PN mapping), and process name-application tree (NA tree) relationships. We first retrieve all the network adapters because, in a Windows environment, there are often multiple network adapters, and the name of the network adapter is inconsistent with the name of the network access. The phenomenon can

confuse the choice of adapters that need to listen to. Therefore, we first query the adapter configured with a gateway because only the adapter which accesses the network will be assigned a gateway on the same network segment as the IP address. Then we obtain the PP mapping from the current ETT and EUT, the PN mapping from the process table, and the NA tree from the prior knowledge base.

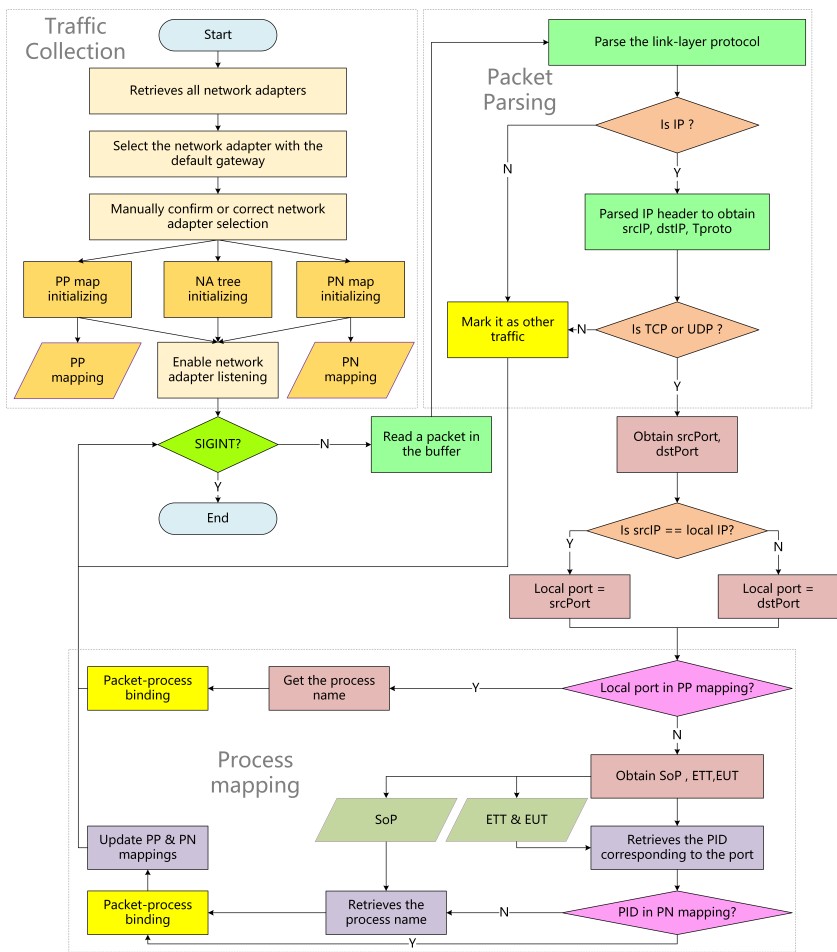

**Figure 4.** A3C-Windows sample collection and construction.

Then, in packet parsing and process mapping, the three mapping relationships are updated in real-time, and the continuous association of "port-PID-process name-application" is realized to form labeled samples automatically.

It is important to note that the traffic could be encrypted on the network or the data link layer. The label is acquired before the data encryption in sample construction, but the sample is acquired after. If the transport layer is encrypted, it is necessary to advance the collection point before the encryption.

Under Windows, a semi-automated sample collection is built to correspond to the regular operation of the user, which is user transparent. Moreover, fully-automated sample collection and construction can be achieved through scripts. Since the user is only responsible for using the application, no manual error is introduced into the collection and labeling process.

### 4.2.2. Linux

As a server-level OS, the Linux platform has the particular condition of persistently high traffic and multiple users. We obtain the "UID-PID" (User IDentifier) tree relationship by monitoring the generation and invocation of the kernel state process in real-time to isolate the same application traffic of different users.

In addition, there are some differences in the system mechanics between Linux and Windows. Linux is an OS that relies entirely on the file system, even though some of the files under the path are virtual and exist in memory. Therefore, to meet the continuous high concurrency under Linux, similar to Windows, we obtain the mapping relationship between Linux in-memory process and network-related information. However, since Linux does not have a direct port-PID mapping, we do the correlation by reading the *inode* intermediate state. Finally, the continuous association of "port-PID-process name-application-user" is realized.

The A3C-Linux sample collection and construction procedure is demonstrated in Figure 5.

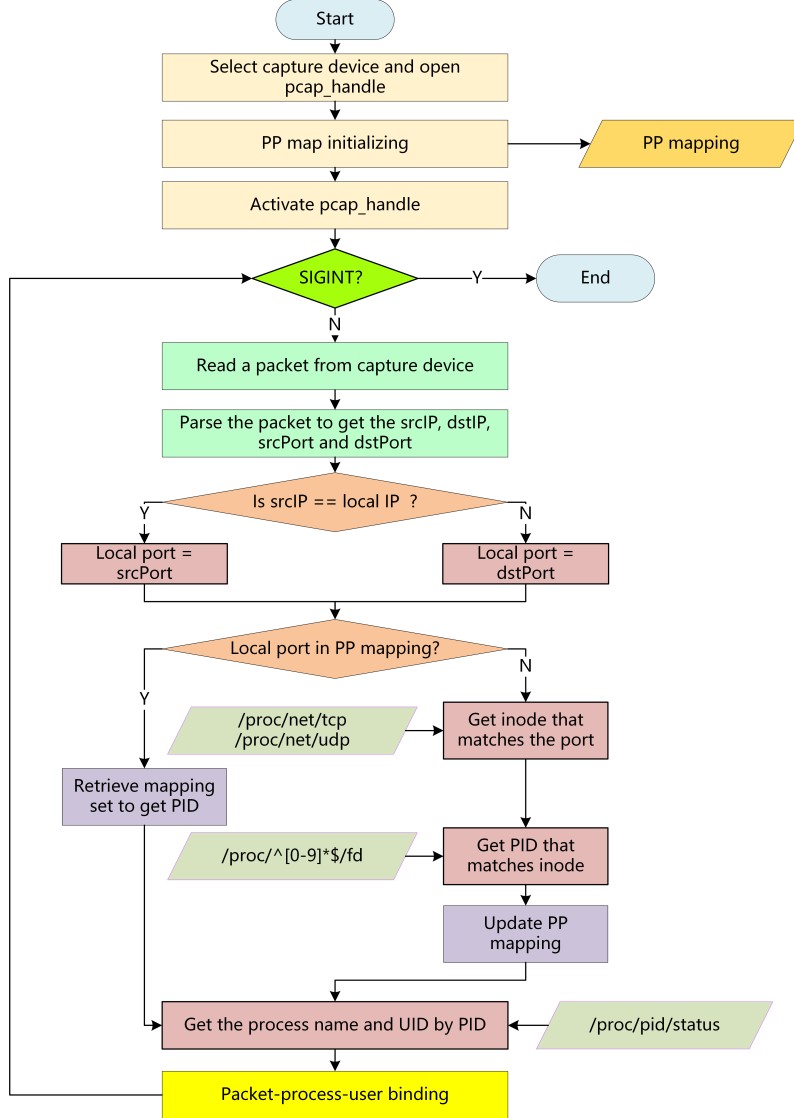

**Figure 5.** A3C-Linux sample collection and construction.

As seen from the flowchart, there is not much difference between A3C-Linux and A3C-Windows during the initialization phase. However, since Linux depends on the file system, the ETT and EUT that should be obtained from the API in Windows need to be fetched from TCP and UDP in the /proc/net path to obtain the inode in Linux. Since the nature of the /proc path is to save all kinds of process-related information, we need to find the PID matching the inode through regular matching to achieve PP mapping.

Like Windows, semi-automated and fully-automated sample collection builds are easy to implement. However, if the device deploying A3C-Linux is a server, the server port

is multiplexed (multiple clients communicate through the same server port). Therefore, bidirectional port numbers must be considered when associating processes with traffic.

### 4.2.3. Android

The current methods to realize traffic collection under the Android mobile devices need to obtain root permission. However, root access is easy to cause irreversible damage to the system and limits the large-scale construction of the experimental bed. Therefore, we propose a method to collect the traffic by using the *VPN Service* interface of Android without obtaining root permission and realizing application filtering. The Android-optimized A3C architecture is shown in Figure 6.

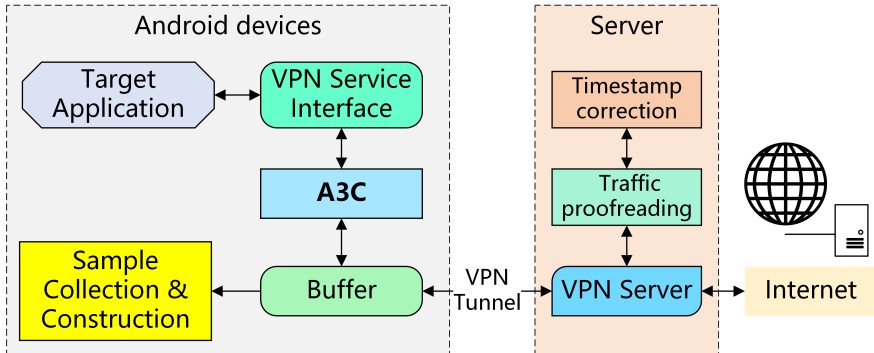

**Figure 6.** A3C-Android sample collection and construction.

The *VPN Service* interface is realized by establishing a VPN communication link to a configured VPN server. However, since A3C-Android is only used for sample collection and construction, the VPN server can be deployed near the end and support multiple A3C-Android instances. It is worth noting that because the VPN Service adopts the TUN mode, the collected labeled samples do not contain Ethernet frame headers.

Since the VPN service interface has been occupied, if an Android device needs to collect application samples in a specific VPN, it needs to use the VPN bridge to collect application samples (the actual VPN client is deployed on the A3C-Android VPN Server). The script is one way to achieve full automation on Android. However, due to the limitations of Android, we propose a method to monitor the changes of UI controls in the foreground of Android devices by using the *Accessibility Service* interface.

### 4.3. Traffic Sample Correlation for Multi-Process Applications

The application classification of encrypted traffic is an essential practical requirement, which requires application-level samples. We looked at most of the top apps and browsers on the PC in 2022 and found very few single-process apps.

Therefore, a relatively complex application may contain multiple processes, and the processes are generated probabilistically during the running of the application. The process generation of complex multi-process applications is mainly sourced by process creation (i.e., open system call) and process expansion (i.e., fork system call). Without considering the restoration of functional processes by daemons, the relationship between processes meets the tree principle. Each process is identified by a PID in the operating system, which records the PID of its parent (either open or fork). Therefore, combined with the network traffic generation mode, the traffic generation mode of the multi-process application is shown in Figure 7.

We first monitor the process activities of the OS to obtain the PID of the current process and its parent process. Then we can dynamically obtain the calling relationship between multiple processes when an application is running. On the other hand, since processes originate from executables in the application installation path, we model the executables and their calling paths in advance to obtain a trusted process tree relationship. Finally,

the relationship of application, process, PID, port, flow is constructed, and the complete application-level samples are obtained.

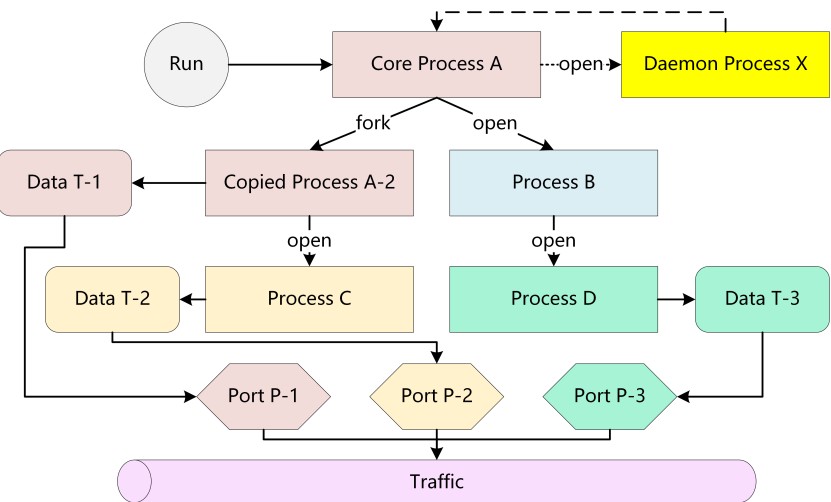

**Figure 7.** Traffic generation and process tree development in multi-process applications.

## 5. Asynchronous Sample Encryption Validation and Context Flow Correlation

After the previous procedure, the current sample data are the segregated core network traffic samples for an application, not the complete application-level encrypted traffic sample. Therefore, to obtain complete application-level encrypted traffic samples, it is necessary to verify the current sample's encryption, authenticity, and context flow integrity. This part corresponds to the offline validation procedure in the A3C system, including the detection module to realize the sample encryption validation, the cross-validation module to realize the TLS sample authenticity validation, and the offline DNS correlation module.

### 5.1. Authenticity Validation of Samples Based on TLS Parameters

TLS is the most mainstream cryptographic communication protocol at present. Because there may be much external link interference (such as advertising) in the current application, not all TLS samples captured, belong to the application. There is a possibility of embedding other applications or external TLS links, so sample authenticity validation is required. Therefore, we propose a TLS encrypted traffic sample authenticity validation method based on TLS handshake parameters. We use parameters such as Server Name Indication (SNI) for reverse validation. Although SNI, as an editable and nullable field, is not suitable for directly encrypted traffic classification, it can validate the authenticity of samples because it contains domain information. It can be used to filter obviously problematic flows. We obtained SNI and other fields from traffic and established a local database to verify it through preliminary research and collection.

### 5.2. Encryption Validation Based on Segmented Entropy Distribution Capsule Neural Network

HEDGE [32] is a method to differentiate encrypted, compressed, and plaintext traffic by using randomness measures. However, stream quantization encrypted files are different from encrypted traffic, which uses a security protocol. It includes a symmetric encryption algorithm, key negotiation, anti-replay, and other mechanisms. Therefore, there will always be plaintext information in encrypted traffic.

Most of the current traffic exists in an encrypted form, but there is still unencrypted traffic. Moreover, there are still many private protocols in the actual network environment, and the highest visible protocol (HVP) [44] of this kind of traffic may be the transport layer protocol. Therefore, it is not possible to directly determine whether the traffic is encrypted or not. Thankfully, the A3C's encrypted traffic detection does not require the consideration of interference with compressed traffic. The reason is that the compressed traffic in the backbone network of the Internet is generated by Internet of Things devices. It has been

found that some Internet of Things devices cannot carry out effective encryption operations due to the limitation of energy and power consumption, so they use compression instead and save bandwidth resources at the same time. On the Internet, compression behavior exists in the application layer protocol. Even some of the compression directly applies to files (such as JPEG format for pictures and AVI format for videos). Therefore, even if the compressed file is transmitted, it is encrypted by the encryption protocol, making the corresponding traffic encrypted. The goal of A3C encrypted traffic detection is to verify the encryption of the sample, and the Internet of Things compressed traffic will not appear in the sample on the Internet side.

In classification, plaintext features are more significant. Some samples of encrypted flow with a low encryption ratio may cause concept drift to classifiers. It is necessary to validate the encryption of application-level network traffic samples obtained in the A3C to further improve the samples' purity.

We introduce deep learning into encrypted traffic detection. Currently, there are three base models of deep learning neural networks: CNN, RNN, attention network, and CapsNet. Since our encrypted traffic detection needs to consider the relationship between the encryption distribution of each data block (the relationship between N-truncation entropy values), we use CapsNet as the base model. Its vector neurons and dynamic routing mechanism can effectively describe the stochastic evolutionary relationship between each block (because the block symmetric encryption algorithm encrypts the network traffic). Therefore, we proposed the Segmented Entropy Distribution Capsule Neural Network (SED-CapsNet) for encryption validation of non-public protocol traffic samples. It is a model that can realize detection with a tiny sample size. The encryption validation procedure of A3C with SED-CapsNet is shown in Algorithm 1.

It is worth noting that we select the 11th packet to be detected in TCP traffic and the 8th packet to be detected in UDP traffic. The selection of these two parameters stems from our research on the handshake process of existing open encryption protocols. In TLS-1.2, which uses TCP as the transmission protocol, the TCP handshake requires three packets, and the TLS handshake requires seven packets. "TCP-TLS(1.2)" is the protocol family that consumes the most packets in the handshake process of known open Internet protocols. Therefore, we skip this step because handshake packets of encryption protocols contain many plaintext information, which may cause considerable misjudgments in encrypted traffic detection.

The network structure of SED-CapsNet is shown in Figure 8.

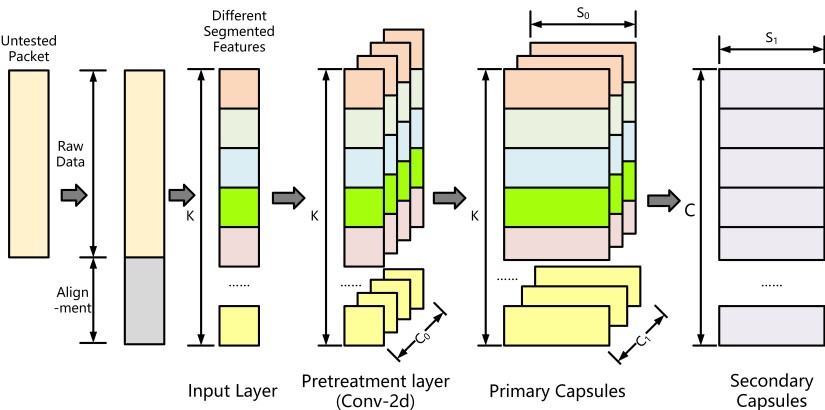

**Figure 8.** Segmented entropy distribution capsule neural network.

---

**Algorithm 1:** Encryption validation procedure of A3C

---

**Input:** Flow *F* or truncated flow $F_t$ or packet sequence *Pkts*
**Output:** Encryption semaphore of input flow *E*, 0 means unencrypted, 1 means
      encrypted
**if** *F.$T_{proto}$ == UDP* **then**
    // Can be added by prior knowledge
    **if** *F.HVP == [QUIC,DTLS,...]* **then**
        *E* = 1
        return *E*
    **end**
    **if** *F.HVP == UDP* **then**
        *Pkt* = *F*[7] // 8th packet
        *E* = SED-CapsNet(*Pkt*)
        return *E*
    **end**
**else**
    // Can be added by prior knowledge
    **if** *F.HVP == [TLS,MMTLS,SSL,SSH,...]* **then**
        *E* = 1
        return *E*
    **end**
    **if** *F.HVP == TCP* **then**
        *Pkt* = *F*[10] // 11th packet
        *E* = SED-CapsNet(*Pkt*)
        return *E*
    **end**
    return -1 // Ignored
**end**

---

The hyper-parameters of SED-CapsNet are demonstrated in Table 1.

**Table 1.** Hyper-parameters of SED-CapsNet.

| Type | Parameters | Values |
|---|---|---|
| Network Parameters | $K$ | 51 |
| | $C_0$ | 256 |
| | $S_0$ | 8 |
| | $C_1$ | 32 |
| | $S_1$ | 16 |
| | $C$ | 2 |
| | $L$ | 3 |
| Training Parameters | Epoch | 250 |
| | Batch size | 10 |

SED-CapsNet takes only one packet of each untested flow (defined by 5-tuples of source/destination IP, source/destination port and transmission protocol) as the input, as the flow has cryptographic integrity. According to the fixed input requirement of the neuron network, we pad the untested packet if its length cannot meet the input size *K*. The feature used in SED-CapsNet is the segmented entropy distribution. Segmented entropy distribution is a method to obtain the entropy sequence of each segment of the current sequence in fixed-length sliding window *W* and fixed-step *P*. However, since there is only a single sequence, N-truncated entropy [52] replaces entropy to improve accuracy. The N-truncated entropy can be expressed by Formula (1) for a sequence whose element values of a given length $N_t$ conform to the uniform distribution U.

$$H_{N_t}(U) = \frac{1}{m^{N_t}} \sum_{n_0 + \dots + n_{m-1} = N_t}$$
$$\left[ \binom{N_t}{n_0, \dots, n_{m-1}} \times \left( -\sum_{i=0}^{m-1} \frac{n_i}{N_t} \log \frac{n_i}{N_t} \right) \right] \tag{1}$$

If the N-truncated entropy estimation $\hat{H}_{N_t}(\omega)$ of the untested sequence $\omega$ is close to 4.87816, the sequence can be considered as a random sequence. In order to detect the encrypted traffic based on segmented entropy distribution, we introduce the capsule neural network model (CapsNet). It uses vector neurons instead of scalar neurons, effectively utilizing the Markov relationship between segmented entropy. Moreover, the CapsNet can effectively reduce the amount of sample data needed for learning.

We first use a 2D-convolutional layer to extend the features in the core classification layers because a single N-truncated entropy is only a floating number. Then, the progressive capsule layers detect whether the packet is encrypted. We use both cross-entropy loss and reconstruction loss as the loss function.

### 5.3. Context DNS Flow Correlation

DNS is a protocol that converts IP addresses and domain names. It is widely used in application protocols for communications based on domain names. DNS is not encrypted in most cases, so DNS traffic analysis is usually required in some management scenarios. Therefore, we propose the context DNS flow correlation module. It is worth noting that this module is optional, which is reflected in Figure 3.

Experimental results show that on the end system, because the port call of DNS is generated by the OS, and the port time cycle of DNS flow is very short, the process port mapping cannot bind DNS traffic to an application process.

In order to satisfy other studies that may require a truly complete data sample containing DNS, the DNS flow should be correlated with a specific domain-based application protocol flow, such as HTTP (HTTPS in encrypted traffic). Therefore, we propose a context DNS flow correlation method based on an asynchronous parameter pair. The correlation with the application protocol flow is established by retrieving the transaction ID field, the associated query, the timestamp, and the response packet.

After finishing the three modules, we can obtain complete application-level or process-level encrypted traffic sample data with high confidence.

## 6. Evaluation

In this section, we display the datasets and experimental settings. In addition, the experimental procedure and results are demonstrated after them. The experimental results are also analyzed in depth.

### 6.1. Datasets and Experimental Settings

To reflect the integrity and effectiveness of the system, we build all the datasets used for experiments by the A3C system.

As A3C can support many different system environments, and we need to experiment on both effect and performance, we first choose a personal device with AMD Ryzen 7 3700X 8-Core Processor CPU, 16 GB memory, and RX550 GPU (referred as environment P). This device is to prove Windows deployability. Then, we choose an Android smartphone device with Qualcomm Snapdragon CPU, 8 GB memory (referred as environment S). This device is to execute Android experiments. We also choose a high-performance workstation with AMD 5950x CPU, 64 GB memory, and GTX 3090 GPU to test our encryption detection performance (referred as environment W).

We conducted experiments for each procedure in the A3C system to verify the effectiveness of each part in the system, and the data between the experiments was correlated to

verify the integrity of the overall system. The application of the data used in the encrypted traffic detection experiment and the corresponding number of flows are shown in Table 2.

**Table 2.** Statistic information on the data used in encrypted traffic detection.

| Application | Encrypted Flow Number | Unencrypted Flow Number |
|:---:|:---:|:---:|
| Aim | 10 | 4 |
| BaiduNetdisk | 863 | 12 |
| Chrome | 5994 | 603 |
| Email | 9 | 4 |
| Facebook | 146 | 9 |
| Firefox | 792 | 19 |
| FTPs | 1 | 2 |
| Hangouts | 142 | 3 |
| Icq | 10 | 1 |
| Msedge | 6219 | 286 |
| Netfix | 54 | 140 |
| QQ | 70 | 126 |
| QQBrowser | 385 | 60 |
| QQMusic | 314 | 112 |
| IQIYI | 26 | 2358 |
| ScpDown | 3 | 0 |
| SFTP | 6 | 0 |
| Skype | 139 | 116 |
| Spotify | 77 | 20 |
| Vimeo | 239 | 18 |
| VoipBuster | 2 | 11 |
| VPN | 13 | 6 |
| WeChat | 15 | 208 |
| Youtube | 243 | 191 |
| Total | 15,772 | 4101 |

### 6.2. Comparison with Existing Automated Sample Construction Methods

6.2.1. Effectiveness Comparison under Windows

At present, there is no research on automated traffic sample construction in academia, but an open-source process-based packet capture tool named OpenQPA [6] has been developed in the Windows environment on GitHub. We compared our A3C system and OpenQPA tool in the actual network environment in the national education network under environment P. A typical result is shown in Table 3.

**Table 3.** Quantitative comparison of traffic sample construction in Windows.

| Estimator | OpenQPA | A3C | Baseline |
|:---:|:---:|:---:|:---:|
| Implementation Rationale | Netstat | ETT and EUT | - |
| Label Granularity | Process | Process/App | - |
| Packet Count | 8463 | 9641 | 9641 |
| Duration (s) | 310 | 311 | 311 |
| Captured Bytes | 8,194,421 | 8,999,105 | 8,999,105 |
| Packet Loss Rate | 12.22% | 0 | - |
| Bandwidth Peak (Mbps) | 28.4 | 28.4 | 28.4 |

We take the Wireshark [5] traffic capture record as the baseline. Although it cannot be used for sample labeling, it can effectively count the number of current packets without packet loss as the baseline for capture. Compared with Wireshark, A3C-Windows has no packet loss when the bandwidth peak is achieved in environment P, while OpenQPA has

a serious packet loss rate of 12.22% (1178 loss in 9641). The result means that A3C can effectively construct traffic samples in any end.

It is worth noting that although the duration of the two methods is different (only one second) from the table, they compare the same data under the same environment. Hence, they meet the standard of quantitative controlled experiments.

### 6.2.2. Performance Comparison under Linux

We conducted experiments on A3C-Linux under environment W. These experiments are mainly about the performance of A3C-Linux, aiming to prove that A3C occupies minimal equipment resources and network resources and is better than the existing methods. We still use Wireshark as a baseline. It is worth noting that Wireshark does not have labeling capabilities, whereas A3C-Linux contains automated labeling, so A3C-Linux does more operations than Wireshark.

We first experimented on the occupation of network resources. Note that network resources in this experiment refer to the resources available to the end system rather than the actual upper limit of network resources. Improper design of the network sniffer tool may cause packet blocking. Therefore, the network sniffer's read-and-write operations on traffic may affect the obtaining of network resources on the end system, which needs to be tested.

We have carried out several experiments on the same file under environment W. We transferred a 569 MB file to a Linux server using SFTP and compared the network speeds of the Linux server in only A3C, only Wireshark, or no capture. The experimental results (average of all the experiments) are shown in Figure 9. The x-axis means the downloading time consumption, while the y-axis represents the size of the already-downloaded file.

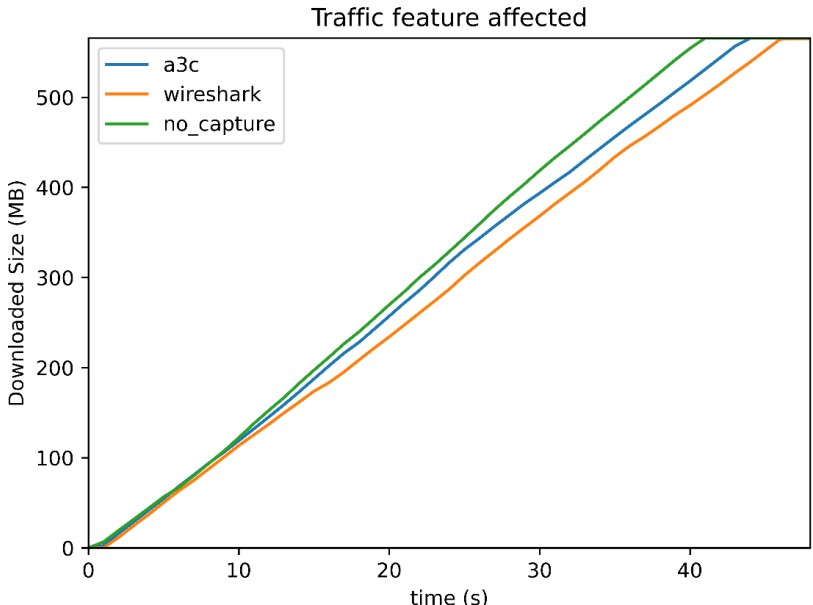

**Figure 9.** Comparison of network resource occupation in three situations.

It can be seen from the results that the transfer rate with A3C enabled is slightly lower than without any capture enabled but higher than with Wireshark. Generally, the transmission speed difference between the three scenarios is insignificant. A3C and Wireshark have little impact on network resource acquisition and do not block packet transmission. The results indicate that A3C does not have any impact on the acquisition of network resources, which is the guarantee of normal service performance.

Then, we conducted experiments on the usage of OS resources. The system resource usage is calculated when only A3C is enabled, only Wireshark is enabled, and neither is enabled. This part of the experiment was carried out in the Linux environment, and a software installation package (about 250 MB) was taken as the sample. The system resource

occupation was monitored during the downloading process. The experimental results are shown in Figure 10.

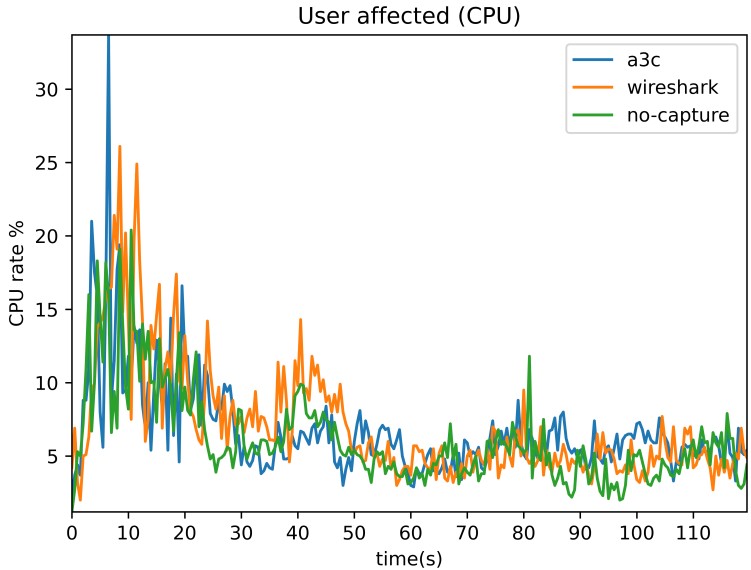

**Figure 10.** System resource occupation comparison of A3C-Linux to Wireshark.

The results show that A3C has a slightly low overall consumption of equipment resources. A3C has a sudden high CPU consumption when it is just started because it needs to apply for some resources from the operating system in advance and conduct PP Mapping initialization. However, with the continuous operation of the program, the resources consumed by A3C are negligible compared with the system resources consumed without capture. The Wireshark consumes more CPU power than A3C to some extent.

6.2.3. Comprehensive Comparison under Android

In the Android environment, there is a public packet capture tool named Packet Capture. The tool utilizes user certificates to capture packets in man-in-the-middle attack mode but requires root permission after SDK24. During the experiment, we connected the Android device to the PC hotspot and enabled the target application and Wireshark on the device. Based on the traffic captured by Wireshark on the PC hotspot, we compared the two methods in environment S.

Firstly, we conducted experiments on Packet Capture. Then we also conducted an experiment on A3C-Android, and one pair of typical experimental results is shown in Tables 4 and 5. It should be noted that to ensure the experiment's authenticity, we did not choose to experiment by replaying the traffic but directly using the real traffic. Since Packet Capture software and A3C-Android in Android devices cannot run simultaneously (there is interference), we conducted two groups of experiments under environment S and compared the two tools based on the collection results Wireshark on VPN Server.

It should be noted that although Wireshark is inefficient and has some errors in sample construction, the A3C-Android VPN server is strictly controlled, so it is feasible to use Wireshark as a baseline for two separate experiments. In addition, although the data used in the two experiments are inconsistent, resulting in their duration being inconsistent, the comparison between Packet Capture and A3C-Android is not strictly quantitative but a semi-quantitative qualitative comparison. However, since this experiment aims to prove the unavailability of Packet Capture tools, the results in the two tables can support this goal.

**Table 4.** Quantitative comparison of Packet Capture to Wireshark.

| Estimator | Wireshark | PacketCapture |
|---|---|---|
| Packet Count | 2254 | 815 (reassembled) |
| Duration (s) | 24.830 | 25.112 |
| Mean Packet Size | 640 | 1662 |
| Captured Bytes | 1,442,474 | 1,354,368 |

**Table 5.** Quantitative comparison of A3C-Android to Wireshark.

| Estimator | Wireshark | A3C-Android |
|---|---|---|
| Packet Count | 1480 | 1491 |
| Duration (s) | 23.423 | 22.418 |
| Mean Packet Size (bytes) | 613 | 562 |
| Captured Bytes | 906,709 | 837,843 |

Packet Capture saves packets according to the session, decrypts the requests at the application layer, and saves the results without using the pcap format. A3C-Android captures raw traffic without the link layer, while Wireshark captures packets with UDP tunnel shells. According to analysis, the more packets captured by the A3C are the ones that failed to be sent, and they are discarded before they reach the PC and can be excluded. However, the decrypted application-layer data captured by Packet Capture is challenging to use. The experiment shows that A3C-Android is the optimal automated sample construction solution on Android.

### 6.3. Comparison with Manual Sample Construction Methods

At present, the major traffic sample construction in the industry is still achieved by manual methods, which have many defects. We compare the A3C system with the manual sample construction method by experiments in Windows.

Firstly, we conducted fifty sample purity experiments in environment P and calculated the ratio of interfering packets after filtering out DNS packets. We collected the target app, Chrome, for nearly 312 seconds each epoch. In manual construction, Wireshark was used to collect all traffic of the network adapter. A3C synchronously collected the traffic of the network adapter and filtered out the pure Chrome traffic. We selected three typical experimental results in environment P, as shown in Table 6.

**Table 6.** A3C vs. manual method in Windows (environment P).

| Index | Captured Pkts | | Irrelevance | Toxic Ratio |
|---|---|---|---|---|
| | A3C | Manual | | |
| 1 | 69,764 | 70,383 | 619 | 0.8% |
| 2 | 59,117 | 59,490 | 373 | 0.6% |
| 3 | 70,130 | 70,769 | 639 | 0.9% |

Experiments show that the proportion of interfering traffic increases with the number of packets captured. The main reason may be that the packet density of the traffic generated by the OS is higher than the average application traffic.

At the same time, to exclude the characteristics of Environment P's own Windows operating system configuration, we re-conducted 69 experiments under another Windows device with similar performance to make statistics on the relationship between toxic ratio changes. We drew the experimental results as a dot-line graph, as shown in Figure 11.

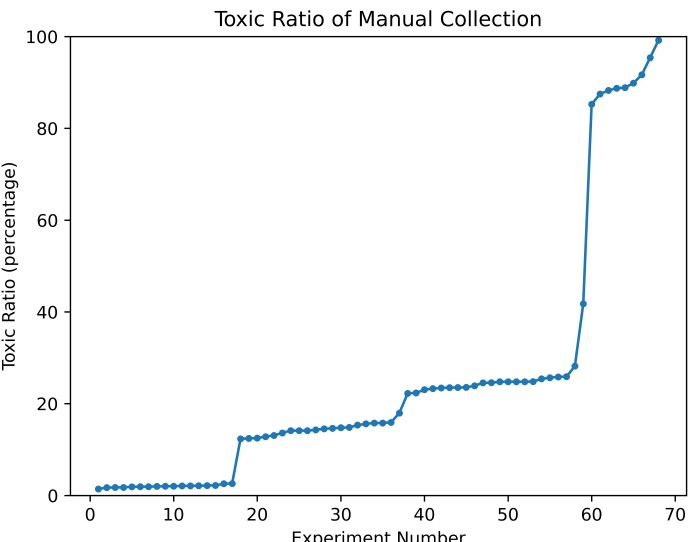

**Figure 11.** Manual method packets toxic ratio in Windows.

The experiment lasted longer, using a script to open ten web pages on the device in sequence. In order to better reflect the distribution relationship of the toxic ratio, our results are ranked from low to high. It can be found that some of the results are consistent with the representative results in the last experiment. The toxic ratio in most of the results was unexpectedly high. The main reason is that the experiment period is very long, in which the Windows operating system may download some Metro application data and update weather and news spontaneously, which will lead to a significant increase in the toxic ratio. In the latter cases, the extremely high level of toxic radio is due to the system update content downloaded by the Windows operating system, which will transmit a large amount of data in a short time, resulting in a sharp increase in the toxic ratio.

Therefore, the blanket sample capture of the manual method is not desirable. There will be some confusion packets and cause devastating sample contamination through system updates at unpredictable times.

Furthermore, we carried out an experiment on the situation of sample construction for multiple applications running simultaneously in environment P. We used Netease Cloud Music, Chrome, WeChat, and QQ concurrently in an experiment. A3C can clearly distinguish the traffic generated by the four applications. However, when the Wireshark is used to collect traffic from multiple applications at the same time, the manual method cannot effectively identify the traffic generated by the application, resulting in low construction efficiency.

*6.4. Experiments of Encrypted Traffic Detection*

As the core of offline validation, the validity of encrypted traffic detection directly determines the authenticity of encrypted traffic samples. If there are some flows with unknown protocols, that is, if the HVP that can be resolved is only TCP or UDP, it is difficult to determine whether the current flow is unencrypted. Unencrypted flow should not interfere with encrypted traffic analysis because there are more direct and definitive analysis schemes. The presence of unencrypted traffic samples affects the authenticity of the samples. Therefore, we compared A3C's encrypted traffic detection module with the current state-of-the-art HEDGE [32] method. HEDGE uses approximate entropy test, block bit frequency test, and cumulative sum test to determine whether a flow is encrypted. The dataset used for encrypted traffic detection and comparison was collected by multiple devices in the national education network environment by the A3C system in December 2021. We conducted several experiments in environment W, each with a sample space of 3000. The average experimental results are shown in Table 7.

**Table 7.** A3C vs. HEDGE in encryption detection.

| Method | Pr | Rc | F1 | Time (s) |
|---|---|---|---|---|
| HEDGE | 90.83% | 64.14% | 75.19% | 2439.28 |
| SED-CapsNet | 87.96% | 67.33% | 83.12% | 368.86 |

It can be seen from the results that the SED-CapsNet in A3C is superior to HEDGE in other indexes, except the precision rate is slightly lower than HEDGE, and the detection time is much faster than HEDGE (only nearly 1/8). The three measures used by HEDGE cost many calculation resources. On the contrary, A3C encrypted traffic detection mainly spends time on feature extraction of segmented entropy distribution, and this part of the cost can be further compressed in engineering implementation.

*6.5. Analysis and Discussion*

From the experiments performed herein, the A3C system performs better in labeled sample collection and construction in both Windows and Android. There are some detailed analyses and explanations of the above experiments, which will be discussed below.

6.5.1. There Are No Collection and Construction Comparison Experiments under Linux

First, there is no valid and publicly available solution under Linux (even though the code, like OpenQPA, is not open source or is directly executable). Using Wireshark to compare functionality with A3C-Linux is redundant, as this has already been done under Windows. The experiment of A3C labeled sample collection and construction on Windows is representative. Furthermore, we put forward seven difficulties in sample collection and label construction of encrypted traffic in Section 1, and not affecting the regular use of users is not reflected in the Windows experiment. Linux is primarily used for servers, and the hardware performance of the Linux server is far superior to that of ordinary personal devices. The performance impact is more pronounced in high concurrency and high-performance scenarios, so we conducted a controlled experiment on Linux concerning performance. The experimental results show that A3C occupies tiny network and OS resources. The actual operation will not feel any change in the application of QoE.

It is worth noting that we did not compare cases of A3C horizontally in three operating systems in the experiment. It is because Windows and Linux systems are mainly oriented to computers (in this scenario, the server is not considered for the moment because the performance and traffic features of the server and the personal computer are quite different, and the result cannot be effectively compared). In contrast, the Android system is oriented to mobile devices. Even if different devices access the same video content in the current Internet environment, the resolution and bit rate are different. Therefore, it is impossible to effectively compare the results of three tools to capture the same content. However, each of the three controlled trials against baseline showed that A3C performed satisfactorily under all three operating systems.

6.5.2. Why Collect the Sample on an Android Device, Not on the Server?

First, it should be made clear that it is feasible to capture traffic in the VPN server of A3C-Android. However, in the actual environment, an A3C-Android VPN server can be used by multiple A3C-Android devices, so it has high consumption to maintain connection and traffic forwarding, and collecting samples on the server may bring performance problems. In addition, the labels are carried out on android devices, and if the samples and labels are not acquired on the same device, there will be problems with timestamp alignment and label information transmission. Last but not least, the current Android devices have adequate hardware performance, which can support local sample collection and preservation, and the mobility of the client makes sample collection on the android devices more rational and comprehensive.

### 6.5.3. Why the Recall of Encryption Detection Is Low?

The recall rate is low because part of the encrypted samples tested as unencrypted. The main reason is that the samples adopted in the experiment are TLS and HTTP samples. In TLS samples, there is a plaintext TLS header, so the overall randomness will be lower if the packet length continues to fall. However, our encrypted traffic detection aims to ensure that the samples in the labeled dataset are all encrypted traffic samples (prevent the classifier from using implicit plaintext features, which would cause colossal over-fitting). A high precision rate can guarantee that almost all samples are encrypted, and the wasted only slightly decreases the collection efficiency.

### 6.5.4. Why HEDGE Is So Slow in Encryption Detection?

HEDGE mainly uses three randomness test methods: block bit frequency test, accumulative sum test, and approximate entropy test, all of which take bits as the test calculation unit. The approximate entropy test has a high time complexity, resulting in a long detection time. On the contrary, N-truncate entropy used in SED-CapsNet takes byte as the calculating unit, and more simple and practical features guarantee the accuracy and significantly reduce the time cost. However, because the capsule neural network is very complex, its detection also consumes time, and the cost is where future research needs to focus.

### 6.5.5. Less than 1% Interfering Packets Really Matter?

The interference here is usually due to background traffic from the OS. However, it is worth noting that it interferes with every type of sample since manual collection can only collect traffic for one application at a time on a device. Therefore, background traffic will interfere with each type of labeled sample, resulting in the accumulation of concept drift, thus reducing accuracy. In the case of enormous samples (such as backbone network boundary), even a slight decrease in accuracy will lead to a significant increase in the number of misclassified samples.

### 6.5.6. A3C-Android Introduces a VPN Server Outside of the End System, Is It Achieved the "Execute on the End System" and "Collection and Labeling Procedure Shall Not Affect the Features of Traffic"?

Due to the characteristics of the Android system, a VPN server is necessary, but this server is transparent to the user who operates A3C-Android. The user does not need to configure the server unless the user wants to build the overall environment in a specific environment independently. So it achieves the "Execute on the end system" requirement. In addition, a VPN server functions as a router or switch to utilize the VPN Service interface of Android. Therefore, the VPN server is only responsible for forwarding, which has no impact on the time. There are many routers or switches in a regular network environment. So it does not affect the features.

### 6.5.7. Can A3C Realize Cyber Security by Helping Secure Network Traffic?

The goal of A3C is to obtain adequate, continuously updated samples of encrypted traffic. Currently, much malicious traffic uses encryption to cover the tracks [16], but the traditional access to malicious traffic must rely on the sandbox environment. Otherwise, there may be interference from benign traffic. With A3C, we can obtain high-quality encrypted malicious traffic samples without interference in a completely controlled real device environment. A3C is essential to protect network traffic because many existing encrypted malicious traffic classification methods cannot operate without novel labeled datasets.

To summarize, the proposed A3C system can be applied to different OS platforms, and the performance of each platform is better than the state-of-the-art methods.

## 7. Conclusions

In order to solve the problem that the public datasets are old and the large-scale labeled sample datasets are difficult to obtain, which leads to the high threshold of the

research and implementation of encrypted traffic classification, we propose a cross-platform encrypted traffic labeled sample dataset collection, construction, and correlation system A3C. The online acquisition procedure in A3C includes the process-level and application-level labeled sample collection and construction methods used in Windows, Linux, and Android. A3C has no packet loss and consumes tiny network and device resources. The offline validation procedure in A3C can effectively detect encrypted traffic and implement TLS authenticity validation and DNS correlation. Overall, the A3C system guarantees the dataset's integrity, purity, authenticity, and credibility. The system solves seven difficult problems in the acquisition of the labeled sample datasets of encrypted traffic. It allows users to build new and effective datasets according to their own needs without relying on old public datasets. The experiments demonstrate that each procedure of A3C is better than the existing solutions.

In future research, we first need to improve the efficiency of encrypted traffic detection. Although asynchronous deployment can improve the efficiency of encrypted traffic detection, experiments show that this overhead is still high. We need to improve the efficiency to achieve near real-time encrypted traffic detection to improve the efficiency of dataset construction. Moreover, there are multiple labeled sample collection points in the actual network environment. The labeled sample sharing between A3C collection points will be a big problem, and this problem can be solved by distributed machine learning solutions such as federated learning. Therefore, the federalization of A3C is also a research focus in the future.

**Author Contributions:** Conceptualization, Z.C. and G.C.; methodology, Z.C. and Z.X.; software, Z.C., Z.X., K.X., Y.S. and J.Z.; validation, Z.C., G.C., Z.X., K.X., Y.S. and J.Z.; formal analysis, Z.C.; investigation, Z.C. and Z.X.; resources, Z.X., K.X. and Y.S.; data curation, Z.X., K.X. and Y.S.; writing—original draft preparation, Z.C.; writing—review and editing, Z.C.; visualization, Z.C.; supervision, G.C.; project administration, G.C.; funding acquisition, G.C. All authors have read and agreed to the published version of the manuscript.

**Funding:** This research was funded by the General Program of the National Natural Science Foundation of China under grant number 62172093.

**Institutional Review Board Statement:** Not Applicable.

**Informed Consent Statement:** Not Applicable.

**Data Availability Statement:** The systems in executable files have been released at: https://data.iptas.edu.cn/web/tbps (accessed on 15 April 2022). An anonymized dataset obtained by A3C has also been published at: https://data.iptas.edu.cn/web/tbps (accessed on 15 April 2022).

**Conflicts of Interest:** The authors declare no conflict of interest. The funders had no role in the design of the study; in the collection, analyses, or interpretation of data; in the writing of the manuscript; or in the decision to publish the results.

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
