# Peer review of "A3C System: One-Stop Automated Encrypted Traffic Labeled Sample Collection, Construction and Correlation in Multi-Systems"

_applsci, doi:10.3390/app122211731_

Round 1

Reviewer 1 Report

Comment and suggestions for Authors

1. there is a problem in the references (no link to the references e.g. reference number). The readability of the first part of the article is comporomised.

2. the first reference is a 2021 ref at the end of the paper ... in the text is reported October 2022 (inconsistence?).

3. The problem is mainly demostreted in fig 1 (Is "demostrated" the right word ')

4. The position of internet is partialy wrong in fig. 1 (the user connect to the cloud through internet)

5. The authors claim to solve seven "difficulties" in with the paper. Two was not solved in Android case. A. Execute on the end system (They introduce a VPN server outside of the end system). B. Collection and labeling procedure shall not affect the features of traffic (The introduction of a VPN server introduces a new network node and impact traffic features. E.g. timing).

6. Why in algorithm 1 is used the 8th packet of the flow for the elaboration in UDP case and 11th packet for TCP case

7. The authors describe the method described in the paper "HEDGE: Efficient Traffic Classification of Encrypted and Compressed Packets" as the "state of art" and compare their results with HEDGE. The cited paper in not the state of art of the sector (only 60 citation today). The authors should compare their result based on SED-CapsNet with various papers for detection of encripted traffic.

8. Why the correlation between DNS and the flows is perfomed only in the HTTPS flows? DNS in not used only for HTTPS

9. The results (tables 2, 3, 4) are quantitative (numbers), but the comparison can be only qualitative since the durations of the experiments are different.

10. For results in table 2, the baseline in wireshark, but the baseline results are not presented, why?

11. A network sniffer (such A3C) does not oppupies network resources, It can only interfere with the network traffic creation processes.

12. The authors do not complete compare the linux results case with Android and Windows case. They explain that linux is similar to windows case. But since the traffic generation is also related to the operating system, the comparison should be performed.

13. Fig 7 seems not coherent with numeric results (a3c and wireshark very similar in the figure). The label of the fig 7 is not appropriate.

14. The traffic of the experiment of fig 8 is not presented.

15. Results in table 5 refers to 3 experiments. Are this number appropriate?

16. The experiment related in table 6 in not well described (e.g. Apps that generated the traffic, the process of labeling: ground truth)

17. Line 210 "However, Sina F et al. [? ] proposed a traffic camouflage method". The paper is not in the references,

However, the topic is an updated topic. It shoud be better addressed.

Reviewer 2 Report

The paper is overall exciting and fits within the scope of this journal. The topic (A3C System: One-stop Automated Encrypted Traffic Labeled Sample Collection, Construction and Correlation in Multi-systems) is impressive.

In the abstract, the authors have used a few abbreviations without expansion.  

The problem is not clear. To identify the problem clearly, the authors have to elaborate the figure 1.

In the contribution, “HEDGE” is used. Although it is explained in subsection 5.2, it should have been expanded before the contribution.

Flowcharts (figures 3 and 4) are not explained properly.

UCIS is written in line 161 but it is not expanded anywhere.

Throughout the paper, the question mark (?) is written as “[?]”. I do not understand why?

Encrypted traffic and malicious traffic analysis considered in this research are investigated using the proposed research. Although specific encryption techniques and attack models or types are not important in this investigation, authors can add these issues to this research for analyzing the results of securing network traffic.

I think that 3C means Collection, Construction, and Correlation. In many places, both 3C and (Collection, Construction, and Correlation) are used. Authors may check this issue to improve the presentation of the paper. 

Anonymized datasets, encrypted traffic datasets, and malicious data traffic are used in this research. Do all encrypted traffic datasets contain malicious data?   

According to the title, Collection, Construction, and Correlation should be considered in this research but the correlation is not clear in sections 4 and 5. It should be explained explicitly with the definitions.

Figures 3, 4, and 5 provide some details about collection and construction but they do not provide direct information on correlation. The authors should explain the impact of the correlations explicitly.   

Round 2

Reviewer 1 Report

The authors reviewed the paper.

Some notes:

1. Line 500 "HEDGE  [32] is a method to differentiate encrypted, compressed, and plaintext files" -> "... plaitext traffic/packets"?

2. The authors should explain why their paper does not consider compressed traffic. Line 228 "In recent years, the academic circle has gradually considered the interference of compressed flow. ..."
